# Exploring the Long-Term Tissue Accumulation and Excretion of 3 nm Cerium Oxide Nanoparticles after Single Dose Administration

**DOI:** 10.3390/antiox12030765

**Published:** 2023-03-21

**Authors:** Lena M. Ernst, Laura Mondragón, Joana Ramis, Muriel F. Gustà, Tetyana Yudina, Eudald Casals, Neus G. Bastús, Guillermo Fernández-Varo, Gregori Casals, Wladimiro Jiménez, Victor Puntes

**Affiliations:** 1Vall d’Hebron Research Institute (VHIR), 08035 Barcelona, Spain; 2Josep Carreras Leukaemia Research Institute (IJC), 08916 Badalona, Spain; 3Institut Català de Nanociència I Nanotecnologia (ICN2), CSIC, The Barcelona Institute of Science and Technology (BIST), Campus UAB, Bellaterra, 08193 Barcelona, Spain; 4Networking Research Centre for Bioengineering, Biomaterials, and Nanomedicine (CIBER-BBN), Instituto de Salud Carlos III, 28029 Madrid, Spain; 5School of Biotechnology and Health Sciences, Wuyi University, Jiangmen 529020, China; 6Service of Biochemistry and Molecular Genetics, Hospital Clinic, Centro de Investigación Biomédica en Red de Enfermedades Hepáticas y Digestivas (CIBERehd), Institut d’Investigacions Biomèdiques August Pi i Sunyer (IDIBAPS), 08036 Barcelona, Spain; 7Department of Fundamental Care and Medical-Surgical Nursing, Universitat de Barcelona, 08007 Barcelona, Spain; 8Departament de Biomedicina, Universitat de Barcelona, 08007 Barcelona, Spain; 9Institució Catalana de Recerca i Estudis Avançats (ICREA), 08010 Barcelona, Spain

**Keywords:** nanoparticle biodistribution, nanopharmacokinetics, nanoceria, NP long-term accumulation, NP excretion, NP dissolution, nanosafety

## Abstract

Nanoparticle (NP) pharmacokinetics significantly differ from traditional small molecule principles. From this emerges the need to create new tools and concepts to harness their full potential and avoid unnecessary risks. Nanoparticle pharmacokinetics strongly depend on size, shape, surface functionalisation, and aggregation state, influencing their biodistribution, accumulation, transformations, and excretion profile, and hence their efficacy and safety. Today, while NP biodistribution and nanoceria biodistribution have been studied often at short times, their long-term accumulation and excretion have rarely been studied. In this work, 3 nm nanoceria at 5.7 mg/kg of body weight was intravenously administrated in a single dose to healthy mice. Biodistribution was measured in the liver, spleen, kidney, lung, brain, lymph nodes, ovary, bone marrow, urine, and faeces at different time points (1, 9, 30, and 100 days). Biodistribution and urinary and faecal excretion were also studied in rats placed in metabolic cages at shorter times. The similarity of results of different NPs in different models is shown as the heterogeneous nanoceria distribution in organs. After the expectable accumulation in the liver and spleen, the concentration of cerium decays exponentially, accounting for about a 50% excretion of cerium from the body in 100 days. Cerium ions, coming from NP dissolution, are most likely excreted via the urinary tract, and ceria nanoparticles accumulated in the liver are most likely excreted via the hepatobiliary route. In addition, nanoceria looks safe and does not damage the target organs. No weight loss or apathy was observed during the course of the experiments.

## 1. Introduction

In the past years, immunometabolism has raised a genuine interest in the scientific community as a tool to control the immune system responses [1]. The immunomodulation concept emerges from noticing that immune cells have different energetic demands depending on the required biological response. In general terms, the faster the energy is required, at higher power, the more free radicals (reactive oxygen species, ROS) are generated, and ATP production is enhanced, as it happens during inflammation [2]. This increased ROS production, enhances the chemical potential of cells to sustain their defensive response and consequent energy consumption in biosynthesis (cytokines and chemokines) and phagocytosis, led by fast and inefficient oxidation of glucose in the cytoplasm (anaerobic glycolysis) [2]. However, this increased ROS production is toxic, causing oxidative stress that results in the oxidation of proteins, lipids, DNA, and ultimately cell death [3]. Therefore, the modulation of immune cell metabolism is emerging as a tool to regulate deranged inflammatory immune responses. This can be achieved either by using antioxidants to control ROS and therefore cellular REDOX potential [4], or by cutting out fuel supply, as in ketogenic diets [5,6] or the use of metformin [7].

Indeed, the need to protect the organism from excess ROS has driven the interest in antioxidants for decades. In 1970, Linus Pauling published a book titled: “Vitamin C and the common cold” [8], describing the protective and anti-inflammatory properties of natural antioxidants. Since oxidative stress is intrinsic when homeostasis is broken, which occurs during disease, numerous epidemiological and mechanistic studies have shown the remarkable benefits of antioxidants in a wide variety of fields, such as auto-immune diseases [9], chronic inflammation [10], neurodegeneration [11], cancer [12], and infections [13]. Unfortunately, the benefits of antioxidants have not been harnessed yet for clinical use, despite the thorough clinical trials since then, which has been attributed to the poor drug-likeness of current available substances (instability, unspecific reactivity, and low bioavailability at the site of action), and consequently their poor pharmacokinetics (PK) profiles [14,15].

In this context, nanotechnology has shown us how mineral antioxidants, such as metal oxide nanoparticles, especially cerium oxide (CeO_2_) nanoparticles (NPs), and nanoceria are safe and powerful anti-inflammatory substances [16,17,18] and seem to overcome these previous antioxidant PK limitations. Nanoceria is well known for its ability to scavenge ROS in biological systems, acting as an effective antioxidant and consequently anti-inflammatory substance [19]. It has been demonstrated to be highly soluble in biological media, present high bioavailability, disperse well in tissues, have therapeutic action at 50 to 250 micrograms/gram, and more importantly, while its chemical specificity is very low, scavenging different types of ROS (and reactive nitrogen species and reactive sulphur species) only in the case when an abnormally high (pathological) concentration of free radicals nanoceria is active, acting as a REDOX buffer [2]. Finally, as an inorganic catalyst, it can be used many times without being consumed, which allows it to work rather permanently at very low doses, offering chronic protection with a single administration. Accordingly, nanoceria efficacy has successfully been tested in a wide variety of disease models, mainly in inflammatory-related pathologies such as cardiac diseases [20], brain ischemia [21,22,23], diabetes [24], retinal dysfunction [25,26], liver inflammation [27,28], cancer [29], anaphylactic shock [30,31,32], acute [33] and chronic glaucoma [34], radiation-induced injuries [35,36], and neurodegeneration [37,38]. This ability of nanoceria to reduce ROS is crucial, because it avoids the direct damage derived from free radicals. In addition, it also has an important immunomodulatory effect, advocating for its use in medicine via its pharmaceutical development [39].

Despite these appealing therapeutic observations, there is still a fundamental open question: nanoceria’s end of life inside the body and in the environment. To some extent, inorganic NPs are supposed to be non-biodegradable and known to accumulate in different tissues and remain there for a long time. It is well known that slow and inefficient clearance rates are displayed by inorganic matter, leading to long-term toxicity concerns due to accumulation. An example is the case of the so-called frustrated phagocytosis where inorganic matter induces chronic inflammation preluding cancer, as in the cases of silicosis or asbestosis, where hundreds of micrometre inorganic particles cause chronic granulomatosis [40]. Even if the inorganic matter causing these diseases is orders of magnitude larger than the currently employed NPs, this raises concerns about NP accumulation, aggregation, and excretion from the organism in extended periods.

Nanoparticle fate belongs to NP PK, the branch of pharmacology that studies drug administration, distribution, metabolization, and excretion. Nanoparticle PK significantly differs from the traditional small molecule PK, and thus emerges the need to create new tools and concepts to assess NP evolution and fate inside the body. Nanoparticle PK strongly depends on size, shape, surface state, and aggregation state. All these features influence nanoceria biodistribution, degradation, and excretion profile, and hence their therapeutic potential and safety.

Today, there are plenty of short-term NP biodistribution studies in the literature, but there is a lack of fate and excretion studies. Excretion is paramount in the development of a medicine. The primary natural excretion pathways from the body are via the liver to faeces, and kidneys to urine. Regarding kidney clearance, several studies show how very small NPs, smaller than 6 nm (in hydrodynamic diameter), can be readily cleared from the body via the urinary tract [41,42]. Note that glomerular filtration not only depends on size, but it also depends on molecule charge [43]. Kobayashi and co-workers [44] showed that glomerular filtration depended on whether the charge was positive or negative, showing enhanced filtration for cationic molecules. Other researchers reported similar observations on NP charge and size effects [42]. Clearance from the liver is more complex, leading to a slower elimination rate than urine excretion. This slow clearance rate is mediated by the hepatobiliary route. The clearance of NPs from the blood into the liver, and from the liver into faeces has been well described in the studies of Pr. Warren Chan [45,46].

In general, as expected, administrated nanoceria ended up accumulating mainly in the liver and spleen (around 90% of the injected dose) with smaller fractions accumulating in the kidneys and lungs, while the hepatobiliary route was the more often proposed excretion route [46]. It is important to understand that when administered, NPs must overcome different biological barriers before reaching their final destination. In short, the first natural barrier that NPs need to cross is the biological fluids, such as blood or lymph, sweat or tear, and the corresponding extracellular matrix, consisting of macromolecules (protein, vitamins), biological molecules (as saccharides) and minerals, that change in different tissues and compartments, and health status [47], which may promote NP aggregation and/or corrosion. The second natural barrier the NPs will encounter, in contrast to small molecules, is the immune system, which evolved to recognize molecular patterns from foreign substances and commensal organisms [48], labelling them with opsonins, and enhancing their uptake by phagocytic cells, such as splenic macrophages, or Kupffer cells in the liver. If NPs are recognized as foreign substances when entering the body, the immune response will reduce their plasma half-life, decreasing their efficiency in targeting different organs. As a third barrier, the microstructure of blood vessels and tissue also strongly determine NP biodistribution. Generally, NPs with hydrodynamic radii smaller than 6 nm are rapidly excreted through the urine while NPs with hydrodynamic radii between 6 and 200 nm are readily filtered in the liver. Larger ones are directly managed by the immune system in blood/lymph, and also tend to accumulate in the liver or other organs such as the lung or spleen [49].

In further detail, to study NP biodistribution, accumulation and excretion, the processes affecting NPs dispersion in the physiological medium have to be considered first [50]: aggregation, protein corona formation, and corrosion. NP aggregation and corrosion will affect NP quality, safety, and efficacy. First, blood and physiological media are highly saline. Hence, if NPs are stabilized by electrostatic repulsion, they will rapidly aggregate as soon as they enter in the highly electrolytic media. As a result, large aggregates (hundreds of nm) form and sediment, and they are not distributed further. These cases are not addressed in this work. Second, proteins in the media will interact with the NP surface. In this case, protein concentration is critical as high protein concentrations will lead to steric stabilization of NPs, while low concentrations will lead to agglomeration and aggregation of NPs and proteins [50]. Finally, NPs can also undergo chemical transformations, as in the common case of corrosion (reactive chemical dissolution), leading to NP dissolution and the corresponding release of NP constituents. These three events usually coexist, and their rates depend on both NP characteristics and medium composition [51].

Regarding the encounter of NPs with the immune system, it is important to note that the immune system can detect molecular and cellular structures with a spatial resolution of a few nm, which allows for detecting molecular patterns expressed in a great variety of pathogens, including viral and bacterial proteins, and bacterial nucleic acid sequences [52], ranging from a few nm to a few hundred nm, as colloidal NPs, which suggests an intense interaction between both [53]. Thus, NPs can be recognized or pass undetected by the immune system. Once detected, they can be tolerated or induce defensive (pro-inflammatory) or anti-inflammatory responses. The different immune responses induced by NPs have been recently reviewed [53]. Basically, poor solubility, large sizes, and cationic charges at the surface or hydrophobic moieties are rather immunogenic, while small, negatively charged, and highly soluble NPs are relatively undetected and tolerated. In general, when needed, surface PEGylation helps to escape from immune detection [54,55]. Ideally, NPs should be designed to evade the immune system, allowing for penetration into the different body tissues unless the NP target is the immune system [53].

Finally, when considering intravenous (IV) administration, which is likely the most employed nanoceria administration route, it is essential to note that the main blood vessels and capillaries in the body have a continuous lining of endothelial cells with pores of 6 nm. In such conditions, small drug molecules, the vast majority of drugs, can diffuse in and out from the blood vessels into the lymph and vice versa, while the passive transport of large objects, such as proteins and NPs, through these pores, is negligible. Besides, fenestrated capillaries found in the intestine and endocrine and exocrine glands present 50–60 nm pores, while discontinuous capillaries, such as those found in the liver, spleen, and bone marrow, display pores ranging from 100 to 1000 nm, which are where NPs are commonly found [56]. Tight junctions deserve special attention, including the blood-brain barrier (BBB), placenta, retinal, and testis barriers, where pores smaller than 1 nm (400 Da) have been reported, preventing NP passive accumulation [57]. It is worth noting here that blood vessel and tissue permeability is altered during the course of diseases, facilitating the passive accumulation of NPs in the disease areas. This altered permeability can increase the concentration of NPs in those tissues by one order of magnitude [58]. This is the case of the enhanced permeability and retention (EPR) effect in solid tumours [59], where defective angiogenesis results in defective blood vessels with large endothelial pores (a few hundred nanometres) nurturing the tumour, which together with the absence of a functional lymphatic drain facilitates NP accumulation in the tumour [60]. Similarly, blood vessels and tissue porosity increase during inflammation, allowing NPs to accumulate in the inflamed area [37]. It is also important to note that functionalization with directing vectors may significantly change the nanoceria distribution, apparently more than size and aggregation state. Thus, nanoceria conjugated to fluorescein isothiocyanate [61] or edaravone [62] has been observed to cross the BBB. Here it is important to note that during neuroinflammation, caused by many brain diseases, together with growing tumours or traumatic brain injury, will allow NPs to permeate the BBB, allowing for the translocation of NPs, as in the case of Amyotrophic Lateral Sclerosis [63].

This study aims to explore 3 nm nanoceria long-term accumulation and excretion in healthy mice after a single IV administration and compare it with the related experiments. We also studied shorter-term accumulation and renal and faecal excretion in model rats, IV, and orally administered with nanoceria.

## 2. Materials and Methods

### 2.1. Model of the Effects of Protein Corona on Nanoceria Internalization and Intracellular Trafficking

The results reported by Mazzolini J. et al. [64] led them to speculate a model in which, in the presence of serum in the environment, the protein corona that formed around the nanoceria contains the proteins involved in cell adhesion and endocytic pathways. Among these adsorbed proteins, transferrin promotes nanoceria internalization through transferrin receptor clathrin-mediated endocytosis, followed by their storage in vesicles and the endosomal compartment. Under these conditions, cell division, viability, and metabolism are preserved, whereas in serum-free media, the absence of a protein corona around nanoceria induces plasma membrane disruption and metabolism changes.

### 2.2. CeO_2_NPs Adsorption by Human Hepatocyte Cancer Cells

In a recent study, we reported that HepG2 cells, a human-derived cell cancer line, can internalize nanoceria [29]. In vitro experiments confirmed the uptake and retention of nanoceria by human hepatocyte cancer cells, mostly in endosome-like bodies. Human hepatocytes were exposed to CeO_2_NPs (10 μg/mL) for 24 h and subjected to TEM analysis. NPs were strongly attached to the outer leaflet of the plasmatic membrane, free in the cytoplasm, and mostly inside numerous endosome-like bodies of diverse morphology.

### 2.3. Nanoceria Synthesis

Nanoceria was synthesized using a wet chemistry method based on the basic precipitation of cerium (III) nitrate hexahydrate (Ce(NO_3_)_3_ • 6 H_2_O) in the presence of sodium citrate (SC). In detail, a TAMAOH solution (50 mL, 81 mM) was added to a 100 mL solution containing Ce(NO_3_)_3_ • 6 H_2_O (30 mM) and SC (60 mM). The final concentration was: 27 mM TMAOH, 10 mM Ce(NO_3_)_3_ • 6 H_2_O, and 20 mM SC. The reaction mixture was left under stirring overnight at room temperature. Later, the mixture was transferred to a three-necked round-bottomed flask (250 mL) and left under refluxing at 100 °C for 4 h. The resulting mixture was a stable, well-dispersed solution of 3 nm nanoceria at a concentration of 1.72 mg/mL CeO_2_.

Before use, the NPs solution was purified with 3 kDa centrifugal filter units (Amicon-Ultra-15 3K, Merck, Germany), and re-suspended in SC 2.2 mM.

### 2.4. Nanoceria Conjugation

#### MSA Conjugation

To prevent aggregation of the NPs in the bloodstream, and to avoid hypotonic shock, nanoceria was conjugated with albumin from mouse serum (MSA, Merck, Germany) in phosphate buffer (PB), 10 mM at 4 °C for 24 h before injection.

### 2.5. Nanoceria Characterization

#### 2.5.1. Bacterial Endotoxin (LAL) Test

Both synthesis and purification of NPs were performed under sterile conditions and with non-pyrogenic material. To ensure safe NPs for animal administration, nanoceria was tested for LPS levels at Echevarne analysis laboratory (Barcelona, Spain).

#### 2.5.2. Transmission Electron Microscopy (TEM)

Nanoceria was visualized using HRES-TEM (Tecnai F20 S/TEM). Ten µL of the as-synthesized solutions were drop-casted onto a carbon coated 200 mesh copper grid and left to dry for at least 24 h in the air at room temperature. The samples’ average size and distribution were measured using Image J Analysis software by counting at least 2000 particles.

#### 2.5.3. UV-Visible Spectra

UV-vis spectra were acquired with a Cary 60 spectrophotometer (Agilent Technologies, Santa Clara, CA, USA) in the 250–800 nm range, using 1.5 mL plastic cuvettes.

#### 2.5.4. Dynamic Light Scattering (DLS) and ζ-Potential

Malvern ZetaSizer Nano ZS (Malvern instruments, Malvern, UK) operating at a light source wavelength of 532 nm and fixed scattering angle of 173° was used to measure NPs hydrodynamic size and ζ-Potential value. Measurements were conducted in 1 cm path cell and 25 °C. Three independent measures were performed.

#### 2.5.5. X-ray Diffraction

XRD diffraction experiments were performed on a Malvern Panalytical Xpert Pro diffractometer, with Cu-Kα X-rays of wavelength (λ) = 1.5406 Å. The patterns were collected in the angle region between 20° and 95° (2θ).

### 2.6. In Vivo Study Design

All experimental procedures were conducted in strict accordance with the European (Directive 2010/63/UE) and Spanish laws and regulations (Real Decreto 53/2013; Generalitat de Catalunya Decret 214/97) on the protection of animals used for experimental and other scientific purposes, approved by the Vall d’Hebron Research Institute (VHIR) Ethical Experimentation Committee, and further validated by the authorized body of Generalitat de Catalunya (ref. n. 11357).

The experimental procedures conducted in Wistar rats were approved and performed according to the criteria of the Investigation and Ethics Committee of the Hospital Clínic Universitari (Barcelona, Spain), and validated by the authorized body of Generalitat de Catalunya (ref. n. 7907).

Animals received humane care according to the criteria outlined in the “Guide for the Care and Use of Laboratory Animals”.

#### 2.6.1. Long-Term Biodistribution in Healthy Mice

For long-term biodistribution study, 30 adult female BALB/C mice (Charles River Laboratories), of 25 g of body weight and 7 months-old at the time of NP administration, were housed 3 to 4 per cage with ad libitum access to food and water during a 12 h light/dark cycle. Mice were randomly divided into four groups. On day 0, four groups of six mice were IV (retro-orbital injection) administered in the ophthalmic venous sinus, under general anaesthesia using Isoflurane (5% for the induction phase and 2% for the maintenance phase), with a single dose of 5.7 mg/kg of body weight (bw) nanoceria conjugated with MSA in PB 10 mM, corresponding to the maximal volume that can be administered IV. Animals were sacrificed at different times after nanoceria injection: 1, 9, 30, and 100 days. The liver, spleen, kidneys, lung, brain, lymph nodes, ovaries, bone marrow, faecal content, and urine (collected in tared Eppendorfs) were collected and stored at −20 °C before elemental Cerium analysis.

#### 2.6.2. Nanoceria Excretion in CCl_4_-Treated Rats

The study was performed in male Wistar rats (Charles-River, Saint Aubin les Elseuf, France) with hepatic fibrosis induced by repetitive CCl_4_ inhalation, as previously described [65]. The rats were fed ad libitum with standard chow and water containing phenobarbital (0.3 g L^−1^) as the drinking fluid. The animals were exposed to CCl_4_ vapor atmosphere twice a week for 16 weeks. Nanoceria (0.1 mg/kg bw) was injected twice a week for two consecutive weeks, starting at the seventh week after beginning CCl_4_ administration. Nanoceria was dispersed in saline solution containing TMAOH ammonium salts (0.8 mM) and was administered as a bolus (500 μL) through the tail vein. After the last dose of nanoceria, the animals were placed in metabolic cages and 24 h urine and faeces measurements were performed at 3, 21, 42, and 56 days after nanoceria administration.

#### 2.6.3. Confocal Imaging of Nanoparticles in Mice Liver

Adult female BALB/cAnNRj mice (Janvier) were 19–22 g and 6–7 weeks old at the time of NP administration. They were housed three to four per cage with ad libitum access to food and water during a 12 h light/dark cycle. Nanoparticles for IV injections were conjugated with MSA in PB buffer 10 mM at 4 °C, 24 h before injection in order to prevent aggregation in the bloodstream.

Experimental procedure—Optical Microscopy of Tissue sections (see Appendix A).

#### 2.6.4. Organ Distribution of Fe_3_O_4_NPs in CCl_4_-Treated Rats

Please see the details of procedure in Appendix A.

#### 2.6.5. Organ Biodistribution after Oral Administration in Healthy Rats

In this study, CeO_2_NP mixed with polyethylene glycol were administered to healthy Wistar rats daily by intragastric gavage administration (3 mg/mL; 10 mg CeO_2_/Kg of bw) for 14 consecutive days (*n* = 3). Nanoceria was administered mixed with PEG as a standard excipient for oral administration, and thus prevented strong nanoceria aggregation once SC was protonated in the acidic pH of the stomach. The cerium concentration was measured by ICP-MS, and the major organs and serum were collected 72 h after the last administration.

### 2.7. Cerium Content Determination

Digestions were carried out in Ethos™ Easy (Millestone, Sorisole, Italy), an advanced microwave digestion system. First, samples were defrosted and mixed in the digestion solution containing one part of concentrated nitric acid and two parts of water. Subsequently, the digestions were performed under a 200 °C cycle for 1.5 h. Thereafter, elemental cerium in tissue were analysed using ICP-MS (7900 ICPMS, Agilent, Santa Clara, CA, USA), in Chemical analysis service (UAB, Barcelona, Spain).

## 3. Results

We have been working with nanoceria in disease models for over 15 years. According to our experience and the current literature, nanoceria accumulates largely (close to 85–95% of the injected dose) in the liver and spleen. Interestingly, this occurs both in healthy and liver-diseased models, as in hepatocellular carcinoma [29] or liver steatosis [66]. Indeed, liver accumulation of NPs has been extensively described in the scientific literature. An extended work can be found in Ref. [45], where the clearance of the NPs from blood to the liver is reported. In all those cases, the administered nanoceria was given at different doses and administration schedules, and although the NPs derived from similar synthesis, they had different formulations (mainly surface state and aggregation state) and ended up in similar places. Here, an important point to keep in mind is that biodistribution inside the organs is heterogeneous. It often seems assumed that NPs are homogeneously distributed and only a fraction of the organs are employed for elemental analysis. However, organs are complex and host different tissues and cell types, and therefore biodistribution inside the organs is heterogeneous. In model cirrhotic rats, injected nanoceria (5 nm aggregated in 30 nm clusters as described in Ref. [66]), was found heterogeneously distributed in the liver, as shown in Figure 1A, indicating that organs have to be homogenized before a fraction is digested for elemental analysis. There is a quite interesting study [45] where the blood clearance mechanisms were examined in relation to the flow dynamic and cellular phenotype in addition to the vessel’s microstructure. Here, the slower blood flow within the organs enhances the possibility of NP retention, which can help predict the accumulation regions inside an organ.

Besides, the NPs can be processed by different types of cells. A clear example is the processing of NPs in the liver either by hepatocytes, described as the main responsible cells during hepatic filtration [67], or Kupffer cells, the resident macrophages in the liver, responsible for the elimination of apoptotic bodies, protein aggregates, and foreign matter [46]. However, we once observed selective capture of NPs by hepatic stellate cells in the mice liver after a single IV injection (Figure 1B). In this work, we employed 50 nm AuNPs so they could be observed in reflectance mode in the confocal microscope without the need for labels at the NP surface, which could affect their biodistribution. This is not expected to be a general trend or applicable to nanoceria, but this selective NP uptake illustrates a most likely widespread event.

Next, the fate of very small 3 nm non-aggregated nanoceria IV administered in healthy mice was assessed. Size is one of the parameters that has been the focus of understanding NPs biodistribution, accumulation, and clearance. As a first observation, it has been postulated that the bigger the NPs are, the narrower their organ distributions [49]. Size may also have a substantial impact on NP integrity. Cerium ions are made insoluble by oxidation at basic pH, and once nanoceria is dispersed at neutral pH, in the REDOX conditions of living systems, its thermodynamic fate is to reduce back and dissolve in the form of Ce^3+^ ions (see the Pourbaix diagram in Suplemmentary Materials [68]. This generally does not happen due to a remarkably high activation energy for cerium oxide crystal dissolution [69]. However, this activation energy decreases with size, ultra-small NPs being prone to dissolution, facilitating biodegradation. Indeed, the curvature radii of NPs with a diameter 30 nm or below become progressively unstable [70]. Therefore, the smaller the NP, the faster its degradation would be. Note that the estimated minimal thermodynamically stable nanoceria crystal size is about 1.9 nm in diameter [69], which leaves 3 nm NPs on the verge of dissolution. Indeed, rapid nanoceria dissolution has been reported inside late endosomes and endolysosomes [71], and at acidic pH during colorimetric detection of analytes [72].

Nanoceria was synthesized in sterile conditions by basic precipitation of cerium nitrate using a classical hydrothermal approach. Sodium citrate (SC) was used as a complexing and stabilizing agent, and TMAOH was used as a base. A stable colloidal solution of ~3 nm non-aggregated naked surface NPs was obtained [25] and described in Figure 2. Note that a precise description of NP features is critical in controlling the interactions at the nano–bio interface [73]. Morphology and size distribution analysed by high-angle annular dark-field scanning transmission microscopy (HAADF-STEM) and High-Resolution transmission electron microscopy (HRES-TEM) (Figure 2A) indicates the formation of non-aggregated 2.5 ± 0.5 nm quasi-spherical nanoceria particles of high uniformity and narrow size distribution (Figure 2A, insert). The X-ray diffraction patterns of the NPs confirm that the sample displays a pure fluorite single-phase (face-cantered cubic, fcc) crystal structure (Figure 2D). The broadness of the diffraction peaks indicates an ultra-small crystal size, where the Scherrer equation confirms the NP size of ~3 nm. The UV-visible spectrum of nanoceria shows a distinct absorption band at 283 nm with a characteristic bandgap of 3.11 eV (the bulk is 3.19 eV) (Figure 2B and inset). This reduced bandgap manifests the oxygen vacancies that nanoceria accumulates at its surface as it decreases in size. These oxygen vacancies determine nanoceria catalytic behaviour (oxygen vacancies are active sites) and nanocrystal stability (the more oxygen vacancies, the closer to dissolution). The zeta potential indicates that after purification and re-dispersion of nanoceria in 2.2 mM SC, particles are negatively charged (−38.9 ± 16.7 mV, at pH = 9.23 and a conductivity of 0.33 mS cm^−1^) (data not shown). Hydrodynamic diameter and stability in simulated physiological media (DMEM, 10% FBS, pH 7.4) were measured by dynamic light scattering (DLS) and shown in Figure 2C, where as-synthesized nanoceria shows monomodal distribution peaking at 3.96 ± 0.89 nm (PDI = 0.35). As expected, the results show nanoceria aggregation within 24 h in DMEM due to surface charge screening because of the high ionic strength of the media. Nanoceria conjugation with albumin prevented NP aggregation, and a slightly increased hydrodynamic diameter, peaking at 6.28 ± 1.6 nm (PDI = 0.47), was observed. Interactions of nanoceria with proteins in physiological media were previously well described [74]. Because of that, we recommend NPs stabilization in the serum before inoculation, murine or rat serum in case of mice and rats, respectively, normally at a final 1 to 10 mg/mL protein concentration, which also prevents rapid renal clearance of NPs < 6 nm.

In order to determine the biodistribution of the nanoceria, 4 groups of 5 mice each were IV administered with 5.7 mg/kg of body weight (bw) single dose of nanoceria and they were euthanized at different times after injection: 1, 9, 30, and 100 days. Although the most employed vein for administration is the lateral tail vein, we here chose retro-orbital injection in the ophthalmic venous sinus in the back of the eye socket. The reason is that administration in the secondary vein network improves the biocompatibility and prevents adverse infusion reactions. Thereafter, ICP-MS was used to determine cerium concentrations in the liver, spleen, kidney, lung, brain, lymph nodes, ovaries, bone marrow, urine, and faeces. During the experiments, all animals showed a healthy weight and appearance, and normal activity without lethargy or apathy after NP administration (see Appendix A).

As expected, the results showed the main accumulation of nanoceria in the liver, and spleen (82%, and 5% of the injected dose (ID), respectively) (Figure 2, Appendix A). Cerium was not detected in the brain and showed extremely low levels in the other organs, although the presence could be observed in the kidneys, lungs, lymph nodes, ovaries, and bone marrow. With time, Ce concentration in the different organs progressively diminished. The most significant elimination was observed in the liver with a reduction from 82% to 40% of ID between day 1 and day 100. The spleen was the second organ with more accumulation, but the clearance rate was slower (see the excretion rates for the different organs in Appendix A). In the kidney, lung, bone marrow, and ovaries there was low accumulation, and slow clearance rates could also be observed. The striking difference between the liver and the rest may indicate that while there exists a ready mechanism for NP excretion in the liver, NP macrophage accumulation in the same liver or other organs leads to much slower excretion. Interestingly, on day 30, cerium concentration increased in the lymph nodes, which may indicate some re-distribution at longer times. This rebound effect is also observed shortly after initial kidney accumulation (W. Jiménez, unpublished observations). Traditionally, the spleen has attracted little attention in pharmacology, because of its minor role in “classical” drug disposition and/or excretion, but this is changing for “new” generation drugs (nanoparticles, recombinant proteins, and monoclonal antibodies) [75]. For example, Demoy et al. demonstrated the pivotal role of spleen marginal zone macrophages in the uptake of polystyrene NPs [76]. We hypothesize then that in our case, a small portion of nanoceria could have been retained in spleen macrophages, which are known to be a slower clearance route, as discussed below.

The results suggest that the main clearance route is hepatobiliary elimination, showing a ≈ 60% decrease of nanoceria in the liver and ≈ 50% inside the body 100 days after injection. This hypothesis is also supported by the observed cerium content in the faecal samples. Interestingly, the excretion is enhanced in the first 24 h after injection, also showing the highest peak of cerium content in the collected faeces. With time, cerium is also detected in urine. Due to nanoceria’s hydrodynamic diameter (Figure 2C), surface charge, and core density, it is highly improbable that particles undergo renal clearance. Besides, in other studies, no significant urine excretion was reported [77]. For this reason, these traces could be ionic cerium coming from nanoceria dissolution. All in all, this suggests elimination by the two main described mechanisms: (i) renal excretion of cerium ions and (ii) hepatobiliary elimination of CeO_2_ NPs.

Despite the reduced number of time points in Figure 3, we fitted the evolution of cerium content inside the liver, or in all measured organs (AMO) vs. time to standard PK models, where small drugs diffuse rather passively through the body following a first-order reaction kinetics that can be described by an exponential decay of cerium concentration with time. We here use one exponential term for one model compartment and two exponential terms for two model compartments (see Suplemmentary Materials) [78,79]. As peripheral distribution is not expected to be significant with NPs, one would expect monomodal exponential decay behaviour. Divergence from this behaviour indicates how the system eliminates cerium in a more active form (Figure 4). In addition, it appears that after the ninth day and ahead, there is a rather monotonous decrease in the cerium concentration, which could be due to a slow excretion of slowly generated cerium ions. On one side, different nanoceria fractions may coexist, and some nanoceria may stay permanently in the body captured inside tissue macrophages or blood monocytes. On the other side, nanoceria should completely dissolve with time in biological conditions regardless of location, accounting for the slow monotonous ceria concentration decrease observed at long times. A physiologically based pharmacokinetic modelling of nanoceria systemic distribution in rats [80] showed similar results, especially the different NP fate depending on admin route, surface, size, and physicochemical properties.

In another study, to compare if biodistribution was different in healthy or diseased pre-clinical models, we set rats in metabolic cages where cerium was also recovered in the urine and faeces during the experimental timeline. The experimental time in the metabolic cages was reduced for animal welfare. This study used Wistar rats with hepatic fibrosis induced by repetitive CCl_4_ inhalation and water containing phenobarbital as the drinking fluid. The animals were injected through the tail vein with nanoceria (0.1 mg/Kg bw) twice a week for two consecutive weeks, starting in the seventh week after the beginning of CCl_4_ administration [66]. After the last dose of nanoceria, animals were placed in metabolic cages to measure cerium concentration for 24 h urine and faeces deposition at 3, 21, 42, and 56 days. As shown in Figure 5, while the cerium faecal content decreases exponentially, a rather monotonous cerium concentration was observed in urine, supporting a slow and progressive nanoceria dissolution into cerium ions and its corresponding renal excretion [81]. As commented before, ultra-small ceria, close to the thermodynamic crystal stability limit, at neutral or acidic pH, is prone to dissolution.

These results are supported by other studies, where after IV injection of NPs with similar size and surface coating they were localized in both Kupffer cells and hepatocytes [82]. Note that both cells have phagocytic capacities, although hepatocytes are directly implicated in the pathway for biliary excretion (NPs found here will be potentially excreted to the bile), whereas Kupffer cells can only eliminate nanoceria via intracellular degradation. Importantly, the Kupffer cells uptake of nanoceria is often accompanied by a mild and transient activation of the immune system [82], calling for a careful examination of the immunogenic properties of designed NPs.

Regarding long-term accumulation of nanoceria, we would like to highlight the studies of Yokel et al. [83,84], where a large dose of 30 nm cubic nanoceria (87 mg/Kg bw) was IV administered to rats, and thereafter an exhaustive accumulation and excretion study was reported on days 1, 7, 30, and 90. In this case, nanoceria was again primarily accumulated in the spleen, liver, and bone marrow. Interestingly, NPs clearance from blood was studied for two weeks and it showed that less than 1% of the nanoceria was removed, and there was a small decrease of particles in any tissue over 90 days. Besides, while we have never observed significant NP in blood a few hours after injection (data not shown), in this study there is a significant cerium blood content after 90 days, which could indicate particle endocytosis by blood cells. This difference among the results is most likely due to the divergence between the physicochemical properties of the used nanoceria (large vs. small size, aggregated vs. non-aggregated, and spherical vs. cubic). Interestingly, although biodistribution and target organs are similar, excretion profiles seem to differ widely. Here, one could hypothesize that NPs that reach the liver and are processed by hepatocytes will be shuttled outside of the body via the hepatobiliary route while NPs that are phagocytosed by resident macrophages, Kupffer cells in the liver, will stay in the macrophages permanently (and if the macrophage dies, another macrophage will engulf the remains) while NPs may slowly dissolve generating Ce^3+^ ions that are excreted via the urinary tract (Figure 6).

We also explored oral administration. In this regard, nanoceria dispersed in a polyethene glycol solution were delivered to healthy Wistar rats daily by intragastric administration (10 mg CeO_2_/Kg bw) for 14 days (*n* = 3). While most of the cerium was found in faeces, some of it entered the body and was observed in the same organs but at different concentrations than in the case of parenteral administration (Figure 7). Here we realised, that although PEG coating, exposure to rat gastrointestinal fluid promotes the rapid dissolution of the NPs into Ce^3+^ ions, which have been reported to act as Ca^2+^, and can be absorbed in the body without long-term accumulation and therefore being excreted through the urine [85]. Dispersion of nanoceria in simulated acidic solution (pH 1.7) showed both NP aggregation due to the protonation of the SC at stomach pH and dissolution of the NPs. If we estimate the average time of residence in the stomach to be about two hours, it seems that after a rapid agglomeration, nearly half of the nanoceria dissolves and the rest remains as aggregates (see Appendix A). Gastric capsules for mice and rat models are difficult to acquire and administrate, thus, we are exploring lipid NP encapsulation for stomach crossing. Besides, the dissolution of particles to ions could explain the elimination through kidneys and its prominent accumulation in the lung, where Ce^3+^ (CeCl_3_) IV administered to mice has been reported to accumulate [81] before renal excretion. There is a case report where a woman accidentally ingested about 1000 mL of a nanoceria-containing solution while working in an industrial setting where nanoceria (aggregated 20–40 nm NPs) was used as a polishing agent [86]. Cerium was detected both in blood and urine. The only symptoms that she presented were transient scattered red petechial, mainly in her face, and decreased plasma coagulation factors. Indeed, anticoagulant properties have been attributed to cerium ions and other rare earth metals [81,87], but not to their nanoscale counterparts, as far as we know.

Finally, despite the numerous evidence, there is still the question if different NPs or different analytical techniques would confirm these generalities of NP biodistribution, including procedures, and experimental and conceptual approximations. Therefore, we employed the CCl_4_-treated rats to observe the biodistribution of 8 nm NPs by magnetic resonance imaging (MRI), a technique displaying a high potential to localize NPs in vivo because of their high spatial resolution. Since nanoceria NPs lack paramagnetic properties, this protocol was performed using 8 nm Fe_3_O_4_NPs, which are somewhat similar, considering size, shape, and surface state, and a similar dose. We obtained sufficient resolution and contrast to assess Fe_3_O_4_NPs-induced signal intensity changes. The most pronounced changes, presented as an increase in signal intensity, were observed in the liver and spleen. These changes were apparent at the first image time point of 30 min after NPs injection (image not shown) and thereafter showed a progressive increase over 90 min (Appendix A), followed by a progressive decrease in signal intensity. Of note was that the brain image analyses revealed no changes during the observational period. Eight weeks after NPs administration the signal intensity was indistinguishable from that obtained in control rats (image not shown) (meaning that the iron concentration was below the limit of detection expected to be at 50% of the administered dose). These findings coincided with those obtained by analysing ICP-MS tissue Ce accumulation following nanoceria IV injection in fibrotic rats [66].

## 4. Conclusions

In conclusion, while it seems universal that a major capture of NPs happens in the liver and spleen, regardless of their difference in dose, morphology, aggregation, and surface state, or if they are taken by immune or filtering cells, the final tissue/cellular distribution and clearance rate from the body seems extremely dependent on those NP characteristics. In this model, NPs that will escape the immune system can be eliminated through the hepatobiliary route, whereas nanoceria captured by circulating monocytes or resident macrophages can dissolve and the cerium ions are excreted through the urine tract. The fact that it is slowly excreted from the body is promising for nanoceria medical development; however, the excretion rates are low and repeated dosing could lead to toxic accumulation [39]. Interestingly, in toxicity studies, no adverse effects were reported at concentrations much higher than the therapeutic ones, as reported after nanoceria oral administration of 1000 mg/Kg bw for 14 days [88], providing some room for repeated administration despite the slow excretion. Accumulation in the spleen also deserves special attention since much slower clearance rates have been observed. This could be related to cell type interaction and to having monodisperse well characterized NPs that can be degraded. It is remarkable how, until now, nanoceria was reported to stay a long time inside the body and not be excreted, and has also been associated with mild pro-inflammatory activation. However, in this work, we demonstrate that non-immunogenic ultra-small highly soluble non-aggregated nanoceria is indeed excretable and biodegradable, paving the progress towards converting nanoceria into an active pharmaceutical ingredient.

All this data and considerations should help to better understand the PK of nanoceria to enable its successful translation to clinical use. Today, few inorganic NPs have reached clinical use: magnetite, Fe_3_O_4_, and amorphous silica, SiO_2_, which is generally regarded as safe (GRAS) by the FDA [89]. Interestingly, both slowly dissolve in vivo into Fe^2+^ ions and salicylic acid, respectively. We expect that many other NPs will join the future pharmacopoeia, where nanoceria will play a significant role.

## Figures and Tables

**Figure 1 antioxidants-12-00765-f001:**
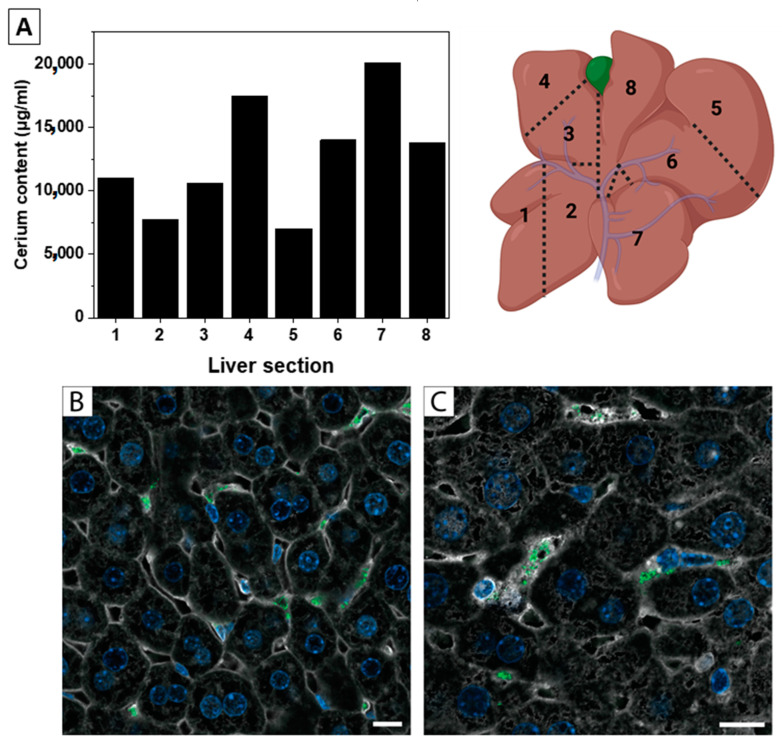
Hererogenicity in organ distribution. (**A**) Liver biodistibution of nanoceria 24 h after IV administration, and liver schema of the digested tissue sections. Liver was divided into 8 parts (schema), and Cerium content of each section is represented in the graph. (**B**,**C**) Confocal images of AuNPs in reflectance mode of a liver thin cut after IV administration of NPs to healthy mice at (**B**) 40× magnification and (**C**) 60× magnification. Actin (white), Hoechst (blue), and NPs (green). Scale bar = 10 µm.

**Figure 2 antioxidants-12-00765-f002:**
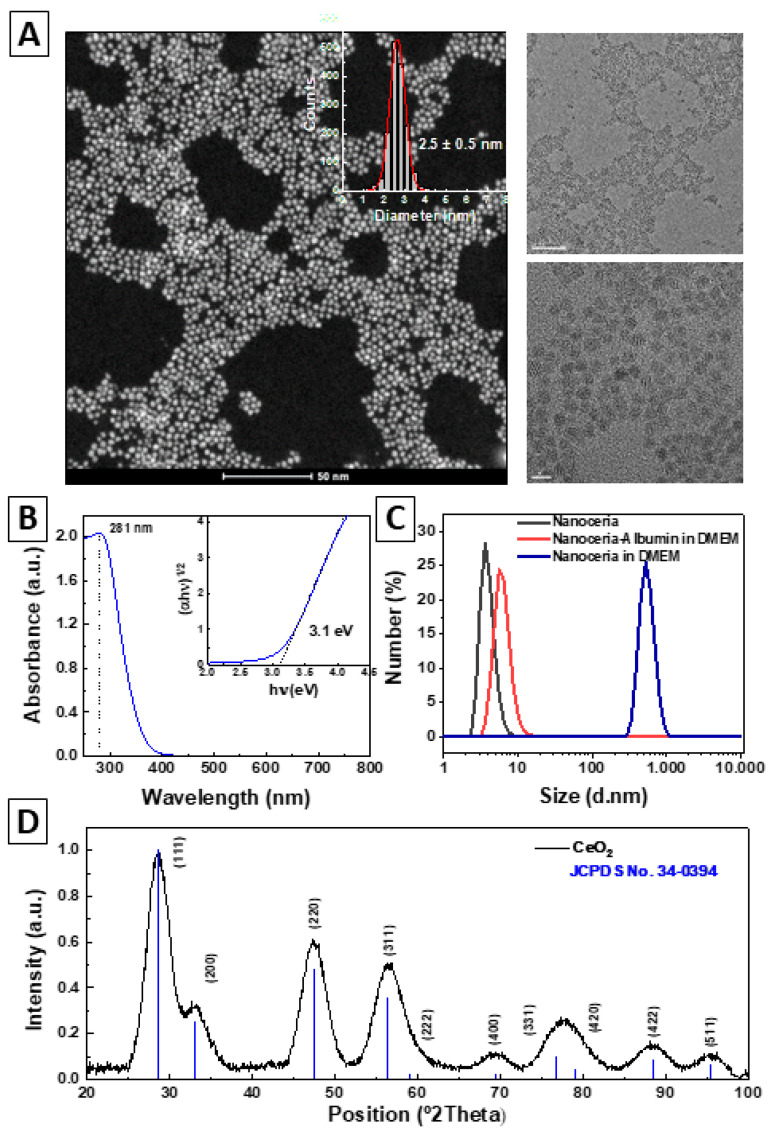
Synthesis and characterization of nanoceria. (**A**) HAADF-STEM images (left) and size distribution (insert), and HRES-TEM images (right). Scale bars are 50, 50, and 5 nm, respectively. (**B**) UV-visible spectrum of the nanoceria and optical bandgap energy determined by Tauc equation (insert). (**C**) The hydrodynamic diameter profile measured by dynamic light scattering of nanoceria (black), nanoceria-albumin in DMEM (red), and nanoceria in DMEM (blue). (**D**) X-ray diffraction pattern of the as-synthesized nanoceria and JCPDS No. 34-0394 standards.

**Figure 3 antioxidants-12-00765-f003:**
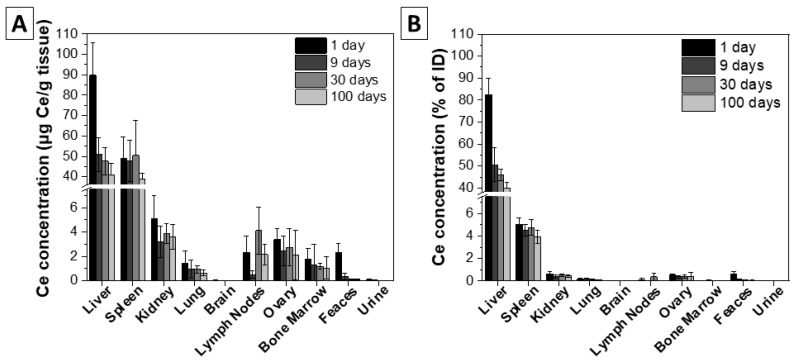
Organ cerium concentration measured by ICP-MS 1, 9, 30, and 100 days after single nanoceria IV administration (5.7 mg/Kg body weight). (**A**) Cerium concentration (µg Ce/g tissue). (**B**) Cerium as a percentage of the injected dose. *n* = 5.

**Figure 4 antioxidants-12-00765-f004:**
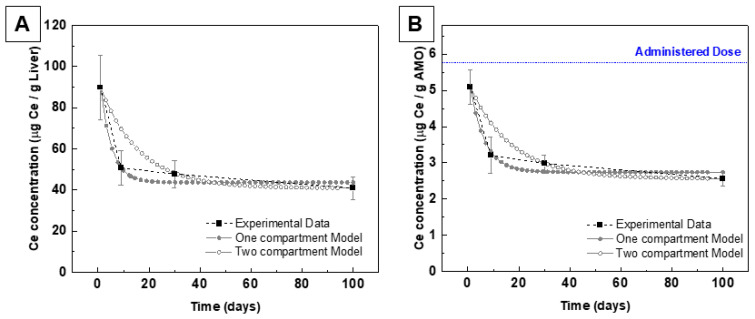
Pharmacokinetics of the CeO_2_ NPs after IV post-injection at 1, 9, 30, and 100 days in the liver (**A**) and in all measured organs (AMO) (**B**). Black squares are experimental data, solid grey circles are the one-compartment model fitting, and the hollow grey circles are the two-compartment model fitting. Blue lines are the administered dose.

**Figure 5 antioxidants-12-00765-f005:**
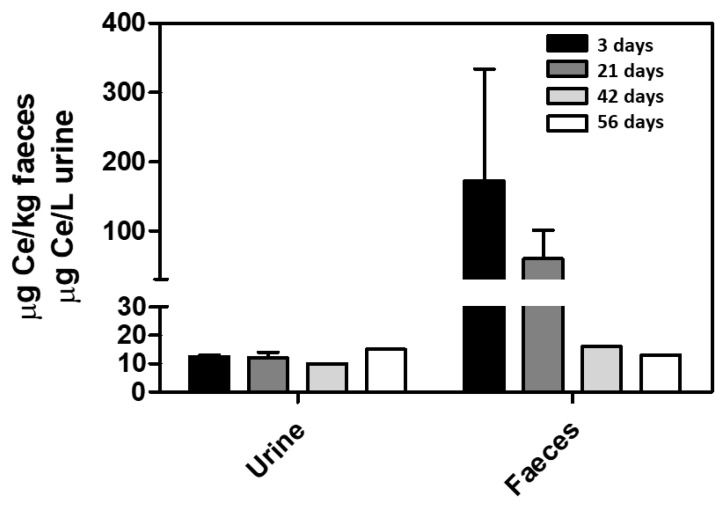
Nanoceria excretion in rats. Cerium concentration was measured by ICP-MS in urine and faeces from CCl_4_-treated rats 3, 21, 42, and 56 days after IV nanoceria administration (0.1 mg/Kg bw) twice weekly for two weeks. *n* = 3.

**Figure 6 antioxidants-12-00765-f006:**
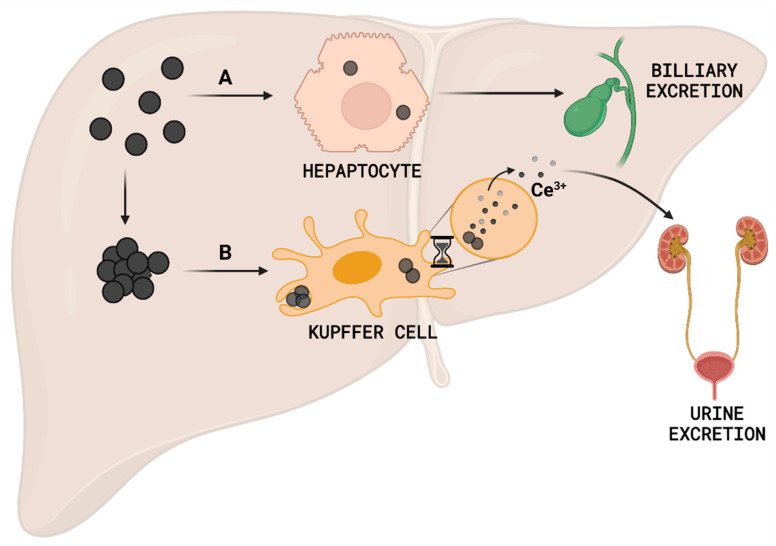
Hypothetical in vivo management of 3 nm nanoceria. After liver accumulation, (**A**) hepatocytes rapidly process non-immunogenic nanoceria to the biliary excretion, (**B**) while nanoceria detected by Kupffer cells will remain inside the cell until it completely dissolves and the produced Ce^3+^ ions are then excreted via the urine.

**Figure 7 antioxidants-12-00765-f007:**
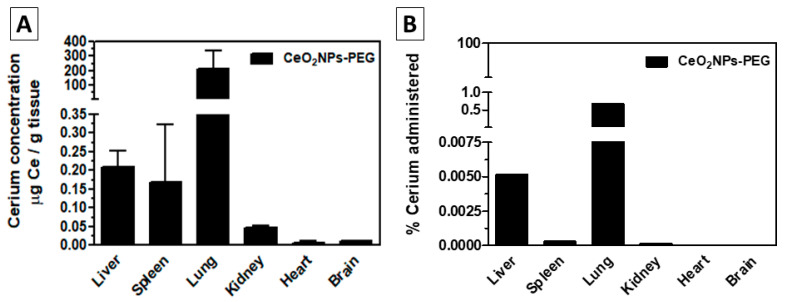
Organ biodistribution after oral administration of CeO_2_-PEG coated nanoparticles (CeO_2_NPs-PEG) in healthy rats. ICP-MS measured cerium concentration. CeO_2_NPs-PEG were delivered daily by intragastric administration (10 mg CeO_2_/Kg bw) for 14 days. Major organs and serum were collected 72 h after the last administration. (**A**) Cerium concentration in µg Ce/g tissue. (**B**) Cerium as a percentage of the administered dose. *n* = 3.

## Data Availability

The data presented in this study are available in the article and Appendix A.

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
