# Peer review of "Exploring the Long-Term Tissue Accumulation and Excretion of 3 nm Cerium Oxide Nanoparticles after Single Dose Administration"

_antioxidants, 2023, doi:10.3390/antiox12030765_

Round 1

Reviewer 1 Report

Currently, cerium oxide (CeO2) is one of the most promising nanobiomaterials holding significant promise in the field of theranostics. The present paper reports on the long-term study of 3 nm CeO2 nanoparticles accumulation and excretion upon intravenous administration. The information obtained on the biodistribution of CeO2 NPs is worth of being published.

I have the following comments:

1. The choice of CeO2 concentrations for IV and IP administration was not justified. What is the reason for administering very high concentrations of cerium oxide which are much higher than typical therapeutic concentrations? Please explain.

2. In Section 2.3.1, there are too many mistakes. This section needs to be completely re-written. Even the title of the Section is incorrect. Please check all chemical formulas, abbreviations, names of the substances, etc.

3. In Section 2.3.1 it is stated that CeO2 was synthesized from cerium(III) nitrate. In Section 3 (line 380), cerium(III) chloride is mentioned. Please check.

4. The information on the optical bandgap of CeO2 NPs (Fig. 2) seems to be irrelevant to the main goal of paper, namely, nano-ceria long-term accumulation and excretion.

Reviewer 2 Report

The current manuscript aims to explore the long-term tissue accumulation and excretion of 3 nm cerium oxide nanoparticles after single dose administration. Although the topic is significant in the field of biocompatibility assessment of metallic nanobiomaterials, there are several issues that definitely require the authors’ attention to improve the quality of this particular manuscript before further consideration for publication in a high-quality journal “Antioxidants”.

Specific comments:

1.         More recently, the authors have demonstrated that topical administration of 3 nm cerium oxide nanoparticles can alleviate AMD in a mouse model (please refer to DOI: 10.1021/acsnano.2c05447 [also reference 22]). This is an extended report aiming to further investigate the safety and fate of administered nanoceria. In my opinion, it may give new insightful viewpoint for clinical translational application of nanobiomaterials. Nevertheless, the authors are highly encouraging to clarify the scientific progress in the academic contribution points between the current manuscript and other earlier reports (please refer to the following examples: #1 DOI: 10.1016/j.nano.2012.08.002 & #2 10.2147/IJN.S157210 & #3 DOI: 10.1002/smll.201907322).

2.         The audiences are unaware of the necessity of checking organ distribution of Fe3O4NPs in CCl4-treated rats. Please specify. Furthermore, this manuscript does not contain any SI file. Please provide this supplementary information (including other necessary experimental procedure and model result) during next file uploading.

3.         The journal readers are curious about why the authors do not unify the animal species for experiments. Some cases involved either Wistar rats or BALB/C mice. Furthermore, both healthy and CCl4-treated animals were used. Whether the results of measured Ce concentration were different attributed to CCl4 treatment?

4.         Please clarify the follow-up time point of data presentation in Figure 1A.

5.         Please clarify the method of calculating Ce concentration in Feaces and Urine. Why the authors do not interpret the data in terms of volume (Figure 3A)?

6.         Please clarify the scientific meaning of unit (μg Ce/g Mice) in y-axis (Figure 4B). Furthermore, as stated by the authors, a single dose of 7.2 mg/kg of body weight nanoceria conjugated with MSA in PB 10 mM, corresponding to the maximal volume that can be administered intravenously. However, according to the blue dotted line (indicative of administered dose) in Figure 4B, the numerical value seems to be below 6 (rather than 7.2). Please carefully check the data presentation again.

7.         The authors should further specify the underlying reason of using CeO2NPs-PEG in Figure 7. It would be better to include the test group without PEGylation for fair scientific comparisons.

8.         As stated by the authors, nanotechnology has shown us how mineral antioxidants, like metal oxide nanoparticles, especially cerium oxide (CeO2) nanoparticles (NPs), nanoceria, are safe and powerful anti-inflammatory substances. However, this important statement was not supported by any documented reference. Over the past few years, some review papers have emphasized the development of CeO2 nanoparticles as anti-inflammatory substances (please refer to the following examples: #1 DOI: 10.1021/acs.chemrev.8b00626 & #2 10.3390/molecules25194559 & #3 DOI: 10.1016/j.cej.2022.134970). The authors are highly recommended to consider the inclusion of these relevant reviews in the reference list to balance scientific viewpoint and update the article content.

9.         As stated by the authors, nanoceria efficacy has successfully been tested in a wide variety of disease models, mainly in inflammatory-related pathologies. In fact, the authors illustrate many diseases, but miss the following important examples i.e., acute glaucoma (please refer to DOI: 10.1016/j.cej.2022.138620) and chronic glaucoma (please refer to DOI: 10.7150/thno.54525). In order to attract more attention from audiences, the authors are highly recommended to consider the inclusion of these publications relevant to nanoceria for inflammatory-related pathologies in the reference list to enrich the research background of the article content.

Round 2

Reviewer 2 Report

The revised version adequately addressed the critiques raised by this reviewer and is now suitable for publication in "Antioxidants".